# A genetically attenuated malaria vaccine candidate based on *P. falciparum b9/slarp* gene-deficient sporozoites

Ben C L van Schaijk[1†], Ivo H J Ploemen[1†‡], Takeshi Annoura[2§], Martijn W Vos[1], Lander Foquet[3], Geert-Jan van Gemert[1], Severine Chevalley-Maurel[2], Marga van de Vegte-Bolmer[1], Mohammed Sajid[2], Jean-Francois Franetich[4,5], Audrey Lorthiois[4,5], Geert Leroux-Roels[3], Philip Meuleman[3], Cornelius C Hermsen[1], Dominique Mazier[4,5,7], Stephen L Hoffman[6], Chris J Janse[2], Shahid M Khan[2], Robert W Sauerwein[1]*

[1]Department of Medical Microbiology, Radboud University Nijmegen Medical Center, Nijmegen, Netherlands; [2]Leiden Malaria Research Group, Parasitology, Leiden University Medical Center, Leiden, Netherlands; [3]Center for Vaccinology, Ghent University and University Hospital, Ghent, Belgium; [4]Centre d'Immunologie et des Maladies Infectieuses, Université Pierre et Marie Curie-Paris 6, Paris, France; [5]Centre d'Immunologie et des Maladies Infectieuses, INSERM, U1135, Paris, Paris, France; [6]Sanaria Inc., Rockville, United States; [7]Service Parasitologie-Mycologie, Assistance Publique—Hôpitaux de Paris, Groupe hospitalier Pitié-Salpêtrière, Paris, France

*For correspondence: Robert. Sauerwein@radboudumc.nl

[†]These authors contributed equally to this work

Present address: [‡]Institute for Translational Vaccinology, Bilthoven, Netherlands; [§]Department of Tropical Medicine, Jikei University School of Medicine, Tokyo, Japan

**Abstract** A highly efficacious pre-erythrocytic stage vaccine would be an important tool for the control and elimination of malaria but is currently unavailable. High-level protection in humans can be achieved by experimental immunization with *Plasmodium falciparum* sporozoites attenuated by radiation or under anti-malarial drug coverage. Immunization with genetically attenuated parasites (GAP) would be an attractive alternative approach. In this study, we present data on safety and protective efficacy using sporozoites with deletions of two genes, that is the newly identified *b9* and *slarp*, which govern independent and critical processes for successful liver-stage development. In the rodent malaria model, PbΔ*b9*Δ*slarp*GAP was completely attenuated showing no breakthrough infections while efficiently inducing high-level protection. The human PfΔ*b9*Δ*slarp*GAP generated without drug resistance markers were infective to human hepatocytes in vitro and to humanized mice engrafted with human hepatocytes in vivo but completely aborted development after infection. These findings support the clinical development of a PfΔ*b9*Δ*slarp*SPZ vaccine.

## Introduction

A vaccine that induces high-level (>90%) sterile protection by inducing immunity that attacks the non-pathologic, asymptomatic pre-erythrocytic stages of *Plasmodium falciparum* (Pf) will prevent infection, disease, and transmission and could be a powerful instrument to eliminate Pf malaria from geographically defined areas (*Plowe et al., 2009*; *malERA Consultative Group on Vaccines, 2011*). In rodent models, sterile protection can be induced by immunization with live *Plasmodium* sporozoites attenuated by either irradiation, genetic modification (GAP), or by concomitant anti-parasitic drug treatment (For reviews see *Hoffman et al., 2010*; *Butler et al., 2012*; *Khan et al., 2012*; *Nganou-Makamdop and Sauerwein, 2013*). In humans, induction of complete sustained protective immunity against a challenge infection has been achieved by previous exposure to the bites of mosquitoes infected with i) live radiation-attenuated *Plasmodium* sporozoites that invade but then completely arrest in the liver

**eLife digest** Vaccines commonly contain a weakened or dead version of a disease-causing microorganism, or its toxins, or surface proteins. These prime the immune system to rapidly recognize, respond to, and eliminate the actual infectious pathogen if later encountered.

While vaccines are currently available to help prevent a large number of diseases, vaccines for many deadly diseases, including malaria, do not yet exist. Malaria is caused by a group of parasites called *Plasmodium*, which are transferred to humans by mosquitoes. While measures to control mosquito populations and prevent mosquito bites have helped to reduce the incidence of malaria in some countries, the number of people—and especially children—that die of malaria every year remains very high.

When a mosquito carrying *Plasmodium* in its salivary glands bites a human, the parasite is injected into the human's bloodstream and travels to the liver. The parasite reproduces in the liver cells until there are so many of them that the cells rupture, and the parasites are released back into the bloodstream. Any mosquito that then feeds on the blood of the infected individual may also suck up the parasite. The parasite then goes through a further stage of development in the mosquito, eventually migrating to the salivary glands, from where the parasite can be transmitted into a new human host.

Recent work in rodents suggests that genetically altered or weakened *Plasmodium falciparum* sporozoites—the form of the parasite found in mosquito saliva—could be used to vaccinate humans against malaria caused by this parasite species. Now, van Schaijk, Ploemen et al. evaluate whether a safe and effective vaccine could be made from sporozoites that lack two genes, called b9 and slarp, which are critical for the parasites to develop inside liver cells. When mice were injected with the modified sporozoites, their immune cells were able to detect the parasites and respond against them. The mice subsequently did not develop malaria when they were infected with normal, unmodified parasites. Furthermore, none of the mice contracted malaria from the modified sporozoites.

The modified sporozoites behaved similarly in human liver cells: after invading these cells, the parasites were unable to develop. Clinical testing and further development are now needed to see if a successful malaria vaccine can be made from these sporozoites.

(*Clyde et al., 1973*; *Hoffman et al., 2002*) and ii) live sporozoites in volunteers taking chloroquine chemoprophylaxis (CPS) with full parasite liver-stage development; once released into the circulation asexual blood stages are killed by chloroquine (*Roestenberg et al., 2009*, *2011*). More recently it has been demonstrated for the first time that sterile immunity can be achieved by intravenous immunization with radiation-attenuated aseptic, purified, cryopreserved Pf sporozoites (SPZ) called PfSPZ Vaccine (*Seder et al., 2013*).

From a product manufacturing perspective, GAPs have the clear advantage of representing a homogeneous parasite population with a defined genetic identity. The genetic attenuation is an irreversible, intrinsic characteristic of the parasite that does not require additional manufacturing steps like irradiation. Furthermore, in the manufacturing process of GAP-infected mosquitoes, operators are never exposed to Pf parasites that can cause disease. However, clinical development of GAPs has suffered from safety problems related to breakthrough infections during immunization leading to pathological blood stage infections responsible for clinical symptoms and complications. Strains of mice showed differential susceptibility to breakthrough infections after injection of sporozoites of rodent malaria GAPs, demonstrating the need for extensive preclinical rodent screening (*Annoura et al., 2012*). The *P. falciparum* GAP PfΔp52Δp36 is the only GAP so far that has been assessed in humans but the trial in which the Pf sporozoites were administered by mosquito bite had to be terminated, because of breakthrough infections in one volunteer during immunization (*Spring et al., 2013*). Our in vitro experiments with PfΔp52Δp36 confirm that this double gene deletion GAP (i.e. two genes removed from the genome) is not fully attenuated similar to the equivalent rodent GAP PbΔp52Δp36 in the *Plasmodium berghei*/C57BL/6 model (*Annoura et al., 2012*). Therefore, identification of additional genes critical and uniquely selective for liver-stage development has become a major challenge for GAP vaccine development (*Annoura et al., 2012*; *Khan et al., 2012*; *Ploemen et al., 2012*). Furthermore, single gene deletion GAPs will most likely not be adequate (*Ploemen et al., 2012*).

This prompted us to generate and test a GAP with deletions of two independent genes critical for liver-stage development. We recently identified a novel *P. berghei* (Pb) gene deletion mutant, Pb$\Delta b9$, lacking the expression of the B9 protein (Pf ortholog: PFC_0750w; PF3D7_0317100) (*Annoura et al., 2014*). This protein is a newly identified member of the *Plasmodium* 6-Cys family of proteins. Initial safety evaluation in rodents demonstrated that Pb$\Delta b9$ mutants have a stronger attenuation phenotype than mutants lacking the 6-Cys proteins P52 and P36 (*van Dijk et al., 2005*; *van Schaijk et al., 2008*; *VanBuskirk et al., 2009*; *Annoura et al., 2014*). As second target gene for liver-stage attenuation, we selected the *slarp* and *sap1* orthologs reported in Pb and *Plasmodium yoelii* (Py), respectively (Pf ortholog: PF11_0480; PF3D7_1147000; hereafter termed *slarp*). These *slarp* mutants show an excellent safety profile by full arrest in the liver in mice (*Aly et al., 2008*; *Silvie et al., 2008*). The SLARP protein is expressed in sporozoites and in early liver-stages and is involved in the regulation of transcription (*Silvie et al., 2008*; *Aly et al., 2011*).

In this study, we report the generation and evaluation of a rodent GAP lacking the genes encoding for B9 and SLARP (Pb$\Delta b9\Delta slarp$) and the generation and evaluation of the equivalent human Pf GAP lacking the Pf ortholog genes. Pf$\Delta b9\Delta slarp$ was generated using constructs that allowed for the removal of the drug selectable marker from the genome by FRT/FLPe recombinase methodology (*van Schaijk et al., 2010*). The safety and efficacy of Pb$\Delta b9\Delta slarp$ and the lack of development of Pf$\Delta b9\Delta slarp$ in human hepatocytes, in vitro, and, in vivo, in chimeric mice provide strong support for clinical development of a Pf$\Delta b9\Delta slarp$ PfSPZ vaccine.

## Results

### Arrest of liver-stage development and induced protection after *P. berghei* $\Delta b9\Delta slarp$ GAP

Previously, we generated a Pb mutant with disruption of the *b9* locus (Pb$\Delta b9$) by standard genetic modification using a double cross-over integration event, followed by removal of the drug-selectable marker cassette by negative selection (*Lin et al., 2011*). Characterization of the Pb$\Delta b9$ phenotype showed that liver-stage development was fully abrogated in BALB/c mice and severely compromised in the more stringent C57BL/6 murine model for *P. berghei* (*Annoura et al., 2014*). Immunization of a single dose of 10k (i.e. 10,000 sporozoites) or 5k Pb$\Delta b9$ protected BALB/c mice against a 10k WT-sporozoite challenge, while 80% of mice were still protected after a single 1k immunizing dose (*Table 1*). In C57BL/6 mice, immunization with 50K/20K/20K of Pb$\Delta b9$ resulted in complete protection lasting up to 180 days, reducing to 45% protection when challenged at 1 year post-immunization.

**Table 1.** Protection of mice after immunization with *P. berghei* Pb$\Delta b9$ or Pb$\Delta b9\Delta slarp$ sporozoites

| Mouse strain | Pb mutant | Day of challenge* | Immunization regimes no. protected/no challenged | | |
|---|---|---|---|---|---|
| **BALB/c** | | | **10k†** | **5k** | **1k** |
| | Pb$\Delta b9$ | 10 | 10/10‡ | 18/20 | 8/10 |
| | Pb$\Delta b9\Delta slarp$ | 10 | 20/20 | 10/10 | 20/20 |
| **C57Bl6** | | | **50/20/20k§** | **10/10/10k** | **1/1/1k** |
| | Pb$\Delta b9$ | 10 | 4/4 | nd | nd |
| | | 90 | 5/5 | | |
| | | 180 | 9/9# | | |
| | | 365 | 5/11 | | |
| | Pb$\Delta b9\Delta slarp$ | 10 | Nd | 10/10 | 6/10 |
| | | 180 | 6/6 | nd | nd |

*Number of days post last immunization; $10^4$ wild-type sporozoites were injected by IV route.
†Immunization dose: number of sporozoites x1000.
‡Protected/total # of immunized mice (%); protection was 0/15 in naive control BALB/c and 0/10 in C57BL/6 mice.
§Immunization dose with 7 day intervals between immunizations.
#Immunization dose 50/10/20k with 7 day intervals between immunizations. nd = not done.

However, sporozoite administration occasionally resulted in blood stage infections after administration of high doses, thereby compromising the safety profile (*Annoura et al., 2014*).

Previously, it has been shown by others that PbΔ*slarp* parasites are completely arrested in liver-stage development with a complete lack of breakthrough blood-stage infections (*Aly et al., 2008*; *Silvie et al., 2008*). Therefore, we generated a new single gene deletion mutant PbΔ*slarp* in a parasite line that constitutively expressed a fusion of the reporter proteins GFP and luciferase, using a *slarp*-targeting DNA-construct for deletion by double cross-over homologous integration (*Figure 1—figure supplement 1*). The PbΔ*slarp* mutant showed blood stage growth and mosquito infections with functional sporozoites similar to wild-type (*Supplementary file 1*). However, intravenous injection of up to 500k PbΔ*slarp* sporozoites never led to full development of parasites in the liver as assayed by in vivo imaging (*Figure 1—figure supplement 1*) or analysis of blood stage infection (*Table 2*). PbΔ*slarp* sporozoites arrested very soon after invasion of cultured Huh7 hepatocytes corroborating the excellent safety findings by *Silvie et al. (2008)*.

Therefore, in order to create a completely attenuated and safe rodent GAP, we additionally disrupted the *slarp* gene in the PbΔ*b9* genome by double cross-over integration (*Figure 1*). Asexual growth and sporogonic development/function equaled wild-type (*Supplementary file 1*). However, PbΔ*b9*Δ*slarp* sporozoites arrested soon after invasion of cultured Huh7 hepatocytes (*Figure 1*) and intravenous injection of 150–200K PbΔ*b9*Δ*slarp* sporozoites never resulted in breakthrough blood-stage infections in mice (*Table 2*). Finally, protective efficacy induced by PbΔ*b9*Δ*slarp* was studied in both BALB/c and C57BL/6 mice. A single immunization dose of 10K, 5K, or even 1K of PbΔ*b9*Δ*slarp* sporozoites in BALB/c mice induced full protection against a 10K wild-type sporozoite challenge (*Table 1*). C57BL/6 mice were 100% protected after 3 × 10K immunization with PbΔ*b9*Δ*slarp* sporozoites, and the protective efficacy reduced to 60% after a 3 × 1K immunization dose. A challenge at day 180 post-immunization of a 50/20/20K dose still resulted in complete protection. The combined data showed that PbΔ*b9*Δ*slarp* completely arrest during liver-stage development and induce a highly efficient protective immunity in two different strains of mice.

## Generation of a *P. falciparum* Δ*b9*Δ*slarp* GAP

Considering the desired phenotype as observed in *P. berghei*, we generated a Pf mutant lacking expression of both B9 (PF3D7_0317100) and SLARP (PF3D7_1147000; sporozoite asparagine-rich protein). These genes are conserved between rodent and human species, both at the level of syntenic location in their respective genomes on chromosomes 3 and 11 respectively, and at the sequence level. Pf*b9* shows 37% amino acid sequence identity and 54% sequence similarity with Pb*b9* ((*Annoura et al., 2014*)); Pf*slarp* shows 28% amino acid sequence identity and 46% sequence similarity with Pb*slarp*.

**Table 2.** Breakthrough blood-stage infections after intravenous injection of PbΔ*slarp* and PbΔ*b9*Δ*slarp* sporozoites

| Mouse strain | Mutant | Infection* Spz x 10³ | Breakthrough blood infection/total # mice | Pre-patent period‡ (days) |
|---|---|---|---|---|
| **BALB/c** | WT† | 10 | 5/5 | 4–5 |
| | PbΔ*slarp* | 50 | 0/5 | |
| | PbΔ*slarp* | 25 | 0/10 | |
| | PbΔ*b9*Δ*slarp* | 25 | 0/10 | |
| **C57BL/6** | WT† | 10 | 5/5 | 4–5 |
| | PbΔ*slarp* | 500 | 0/5 | |
| | PbΔ*slarp* | 200 | 0/10 | |
| | PbΔ*b9*Δ*slarp* | 200 | 0/10 | |
| | PbΔ*b9*Δ*slarp* | 150 | 0/5 | |

*Inoculation dose of sporozoites administered IV.
†*P. berghei* ANKA strain: line cl15cy1.
‡Day with parasitemia of 0.5–2%.

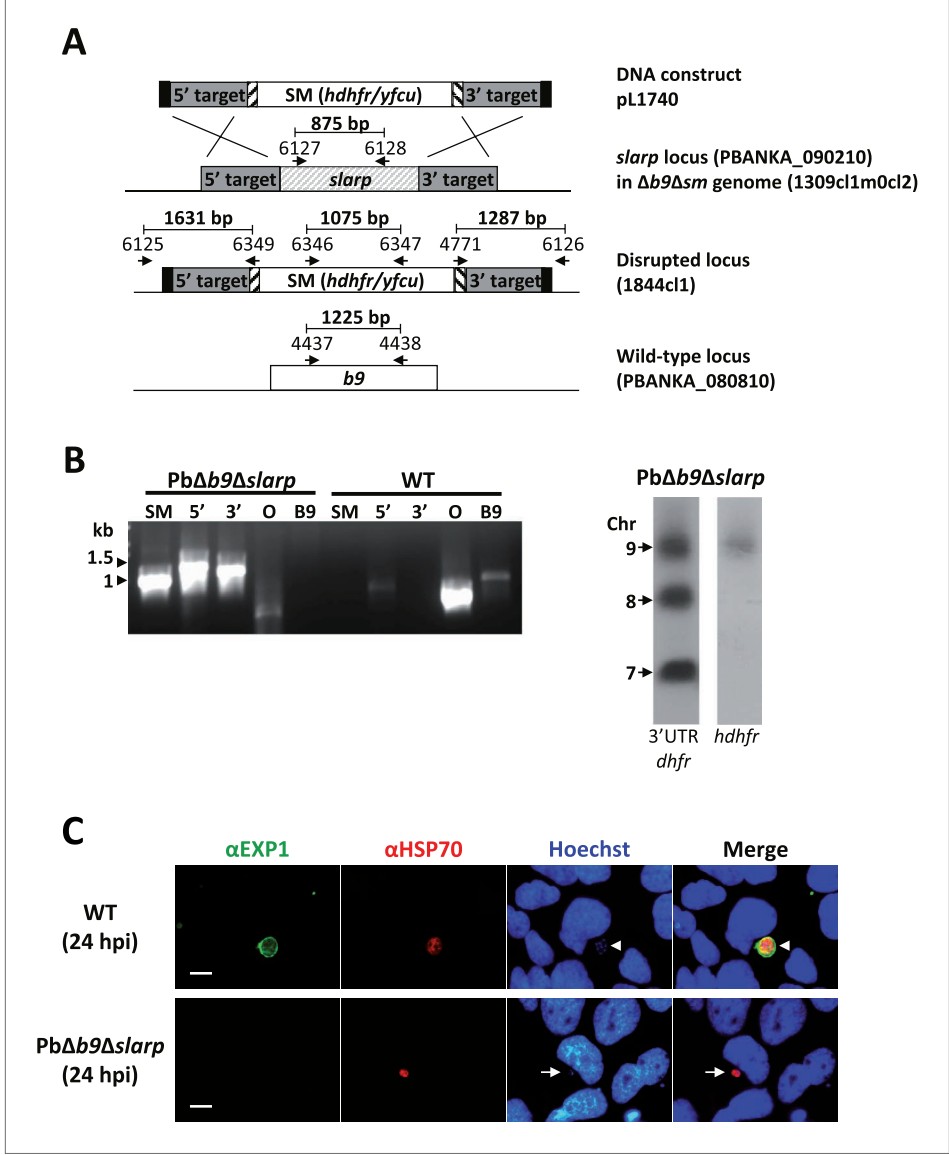

**Figure 1**. Generation and genotype analyses of *P. berghei* mutant PbΔ*b9*Δ*slarp*. (**A**) Generation of mutant PbΔ*b9*Δ*slarp*. For PbΔ*b9*Δ*slarp*, the DNA-construct pL1740 was generated containing the positive/negative selectable marker cassette *hdhfr/yfcy*. This construct was subsequently used to generate the mutant PbΔ*b9*Δ*slarp* in the PbΔ*b9*Δ*sm* mutant. See ***Supplementary file 2A*** for the sequence of the primers. (**B**) Diagnostic PCR and Southern analysis of Pulse Field Gel (PFG)-separated chromosomes of mutant PbΔ*b9*Δ*slarp* confirming correct disruption of the *slarp* and the *b9* locus. See ***Supplementary file 2A*** for the sequence of the primers used for the selectable marker gene (SM); 5'-integration event (5'); 3'-integration event (3'); and the *slarp* and the *b9* ORF. For Southern analysis, PFG-separated chromosomes were hybridized using a 3'UTR *pbdhfr* probe that recognizes the construct integrated into *P. berghei slarp* locus on chromosome 9, the endogenous locus of *dhfr/ts* on chromosome 7, and a 3'UTR *pbdhfr* probe that recognizes the construct integrated into *P. berghei b9* locus on chromosome 8. In addition, the chromosomes were hybridized with the *hdhfr* probe recognizing the integrated construct into the *slarp* locus on chromosome 9. (**C**) Development of liver-stages in cultured hepatocytes as visualized by staining with antibodies recognizing the parasitophorous vacuole membrane (anti-EXP1; green) and the parasite cytoplasm (anti-HSP70; red). Nuclei are stained with Hoechst-33342. Hpi: hours post-infection. Scale bar represents 10 µm.

The following figure supplement is available for figure 1:

**Figure supplement 1**. Generation and genotype analyses of *P. berghei* mutant PbΔ*slarp*-a.

First, we generated two independent Pf mutants lacking *slarp* by standard double cross-over integration of a DNA construct and analyzed their phenotype throughout the parasite life cycle (*Figure 2—figure supplement 1,2*). Blood-stage development of two independently derived PfΔ*slarp* (i.e. PfΔ*slarp-a* and–*b*) parasites was comparable to WT parasites. PfΔ*slarp* mutants produced WT numbers of gametocytes, oocysts and sporozoites (*Figure 2*). The intra-cellular PfΔ*slarp-a* and -*b* parasite development in primary human hepatocytes was not significantly different in number and morphologically identical to WT parasites at 3 and 24 hours post-infection (hpi) (*Figure 3*). However, their number was more than 10-fold reduced at 48 hpi and not detectable from day 3 onwards to day 7 post-infection. Parasites lacking *b9* in *P. falciparum* arrested before day 2 post-infection of primary human hepatocytes with the exception of one observed liver schizont at a later timepoint (*Annoura et al., 2014*). PfΔ*slarp-a* and -*b* parasites still showed positive HSP70 staining and morphologically normal parasites at 48 hpi in primary human hepatocytes, indicating time point of arrest later compared to PfΔ*b9* parasites.

Next, we generated double gene-deletion PfΔ*b9*Δ*slarp* mutants using the FRT/FLPe recombinase methodology (*van Schaijk et al., 2010*). This methodology employs FLPe recombinase to remove a FRT-site flanked drug resistance marker cassette introduced into the Pf genome when the target gene has been removed by double cross-over homologous recombination as shown for PfΔ*slarp-b* parasites in *Figure 2—figure supplement 1,2*. After cloning, this 'marker-free' line was subsequently transfected with the Pf*b9* gene-targeting construct pHHT-FRT-GFP-*b9* (*Annoura et al., 2014*) to delete the *b9* locus from the PfΔ*slarp-b* genome (*Figure 2—figure supplement 1,2*). Subsequently two 'marker-free' clones, PfΔ*b9*Δ*slarp*-F7 and PfΔ*b9*Δ*slarp*-G9, were obtained containing the correct genotype that is removal of the *slarp* and *b9* gene loci as well as both respective drug selection cassettes (*Figure 2— figure supplement 2*). In addition, we confirmed the loss of expression of both *slarp* and *b9* by RT-PCR analysis by demonstrating the absence of transcripts in mRNA collected from PfΔ*b9*Δ*slarp*-F7 and PfΔ*b9*Δ*slarp*-G9 salivary gland sporozoites (*Figure 2—figure supplement 2*). We then examined the phenotype of PfΔ*b9*Δ*slarp*-F7 and PfΔ*b9*Δ*slarp*-G9 mutants during blood stage and mosquito development. Asexual blood stage growth of PfΔ*b9*Δ*slarp* parasites was normal as both clones reached an asexual parasitemia between 0.5 and 5% during cloning within 21 days and PfΔ*b9*Δ*slarp* clones produced WT-like numbers of gametocytes, oocysts, and sporozoites (*Figure 2*).

## Developmental arrest of PfΔ*b9*Δ*slarp* GAPs in human hepatocytes

We next analyzed the development of PfΔ*b9*Δ*slarp* in human hepatocytes using cultured primary hepatocytes and uPA$^{+/+}$-SCID mice engrafted with human hepatocytes (human liver-uPA-SCID mice) (*Meuleman et al., 2005*). PfΔ*b9*Δ*slarp* sporozoites showed normal gliding motility, hepatic cell traversal (*Figure 2*), as well as invasion of primary human hepatocytes, but parasites were completely absent in two independent experiments at day 2 up to day 7 post-infection, following inoculation of primary human hepatocytes with 40,000 PfΔ*b9*Δ*slarp* F7 or G9 sporozoites (*Figure 3*). Detailed analyses of 80 individual wells at day 4 post-infection did not result in identification of a single developing parasite. The combined day 2 and day 4 data of PfΔ*b9*Δ*slarp* indicated that the timing of arrest is similar to PfΔ*b9* (*Annoura et al., 2014*) and there had been complete arrest of liver-stage development, similar to PfΔ*slarp* parasites.

In addition, human liver-uPA-SCID mice were intravenously inoculated with $1 \times 10^6$ WT or PfΔ*b9*Δ*slarp* sporozoites. Two heterozygous uPA$^{+/-}$-SCID mice, not engrafted with human hepatocytes, served as controls and were also challenged with *P. falciparum* sporozoites. Livers were collected either at 24 hpi or 5 days post-infection for detection of *P. falciparum* 18S DNA by quantitative real-time PCR (*Foquet et al., 2013*). Both mice infected with WT Pf and 1 of the 2 mice infected with PfΔ*b9*Δ*slarp* were positive for Pf 18S DNA at 24 hr post-infection, demonstrating successful sporozoite infection in human hepatocytes (*Figure 3*). A lower signal was observed in PfΔ*b9*Δ*slarp*-infected mice at day 1 after infection compared to WT parasites, likely reflecting the early time point of arrest of this GAP. All mice infected with Pf WT (3/3) showed a strong increase in parasite 18S DNA at day 5 post-infection, representing successful liver-stage development. In contrast, none of the human liver-uPA-SCID mice infected with PfΔ*b9*Δ*slarp* sporozoites showed 18S DNA higher than heterozygous uPA$^{+/-}$-SCID mice, not engrafted with human hepatocytes that had been infected with PfΔ*b9*Δ*slarp* sporozoites (*Figure 3*). Although these studies were performed with a limited number of mice, these findings indicate that PfΔ*b9*Δ*slarp* parasites can invade but do not develop in livers of humanized mice. Our combined results demonstrate abrogation of development of PfΔ*b9*Δ*slarp* inside human hepatocytes.

## A

**Phenotypes of *P. falciparum* PfΔ*slarp* and PfΔb9Δ*slarp* parasites.**

| Parasite | Gametocyte stage II (range) | Gametocyte stage IV-V (range) | Exfl. | Oocyst production (IQR) | % Infected mosquitoes (Infected/dissected) | Mean no. of sporozoites (std) |
|---|---|---|---|---|---|---|
| WT | 6,6 (1–13) | 51 (29–61) | ++ | 27 (12–42) | 95% (104/110) | 48 (23–95) |
| PfΔ*slarp*-a | 8,8 (3–15) | 58 (41–65) | ++ | 23 (8–59) | 93% (37–40) | 50 (22–97) |
| PfΔ*slarp*-b | 9,1 (2–27) | 49 (27–70) | ++ | 36 (17–59) | 96% (106/110) | 77 (22–174) |
| PfΔ*b9*Δ*slarp*-F7 | 8,2 (1–140) | 39 (22–47) | ++ | 34 (20–54) | 96% (77/80) | 81 (33–106) |
| PfΔ*b9*Δ*slarp*-G9 | 8,6 (4–12) | 49 (27–65) | ++ | 33 (13–64) | 94% (75/80) | 62 (22–105) |

## B

**Gliding motility**

## C

**Hepatocyte traversal**

**Figure 2**. Phenotypes of *P. falciparum* PfΔ*slarp* and PfΔ*b9*Δ*slarp* parasites. (**A**) Gametocyte, oocyst, and sporozoite production. Gametocyte numbers (stage II and IV–V) per 1000 erythrocytes at day 8 and day 14 after the start of gametocyte cultures. Exflagellation (Exfl) of male gametocytes in stimulated samples from day 14 cultures (++ score = >10 exflagellation centers per microscope field at 400× magnification). Median number of oocysts at day 7, IQR is the inter quartile range and sporozoite (day 21) production (×1000) in *A. stephensi* mosquitoes. (**B**) Gliding motility of *P. falciparum* WT (cytochalasin D treated and untreated), PfΔ*slarp*-b, PfΔ*b9*Δ*slarp*-F7, and PfΔ*b9*Δ*slarp*-G9 parasites. Gliding motility was quantified by determining the mean percentage ± standard deviation of parasites that exhibited gliding motility by producing characteristic CSP trails (≥1 circles) or parasites that did not produce CSP trails (0 circles). (**C**) Cell traversal ability of *P. falciparum* NF54, PfΔ*slarp*-b and PfΔ*b9*Δ*slarp*-F7 sporozoites as determined by FACS counting of Dextran positive Huh7 cells. Shown is the mean percentage ±standard deviation of FITC positive cells. Dextran control (control): hepatocytes cultured in the presence of Dextran but without the addition of sporozoites.

The following figure supplements are available for figure 2:

**Figure supplement 1**. Consecutive gene deletion of *slarp* and *b9* in *P. falciparum*.

**Figure supplement 2**. Genotype analysis of the generated PfΔ*slarp* and PfΔ*b9*Δ*slarp* parasites.

## Discussion

The Pf GAP PfΔ*b9*Δ*slarp* containing two gene deletions is proposed as a whole-parasite malaria vaccine candidate. Rationale and arguments are based on in vitro and in vivo experiments and supported by safety and protection data with rodent Pb GAP with deletions of the orthologous genes.

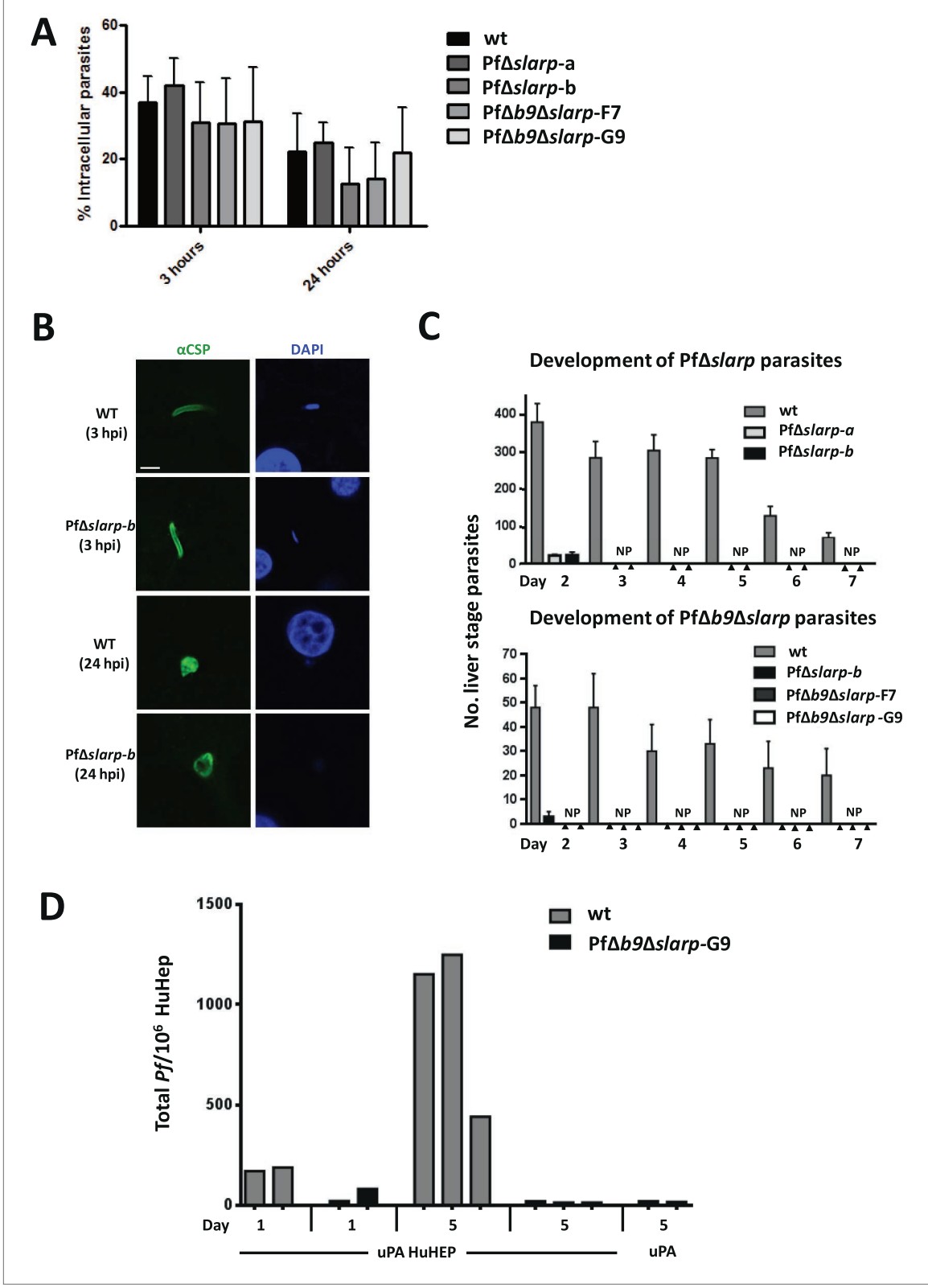

**Figure 3**. Development of *P. falciparum* PfΔ*slarp* and PfΔ*b9*Δ*slarp* parasites in human primary hepatocytes. (**A**) In vitro invasion of *P. falciparum* wt, PfΔ*slarp*-a, PfΔ*slarp*-b, PfΔ*b9*Δ*slarp*-F7, and PfΔ*b9*Δ*slarp*-G9 sporozoites in primary human hepatocytes. Invasion is represented as the mean ratio ± standard deviation of extra- and intra-cellular sporozoites by double staining at 3 and 24 hr post-infection, determined after three wash steps to
*Figure 3. Continued on next page*

*Figure 3. Continued*

remove sporozoites in suspension. (**B**) Immunofluorescence assay of PfΔ*slarp*-b parasites in human primary hepatocytes at 3 and 24 hr post-infection. Parasites are visualized by staining with anti-PfCSP antibodies (green; Alexa-488) and parasite, and hepatocyte nuclei are stained with DAPI (blue). Images were photographed on an Olympus FV1000 confocal microscope. Scale bar represents 5 µm. (**C**) Development of *P. falciparum* wt, PfΔ*slarp*-a, PfΔ*slarp*-b (top panel), PfΔ*b9*Δ*slarp*-F7, and PfΔ*b9*Δ*slarp*-G9 (bottom panel) liver-stages in primary human hepatocytes following inoculation with 40,000 sporozoites. From day 2 to 7, the mean number ± standard deviation of parasites per 96-well was determined by counting parasites stained with anti-*P. falciparum* HSP70 antibodies. The bottom panel represents experiments performed in primary human hepatocytes from 2 different donors. No parasites present (NP). (**D**) Development of liver-stages of PfΔ*b9*Δ*slarp* GAP in chimeric mice engrafted with human hepatocytes. Mice were infected with $10^6$ wt or PfΔ*b9*Δ*slarp*-G9 sporozoites by intravenous inoculation. At 24 hr or at 5 days after sporozoite infection, livers were collected from the mice and the presence of parasites determined by qPCR of the parasite-specific 18S DNA. uPA HuHEP; chimeric homozygous uPA$^{+/+}$-SCID mice engrafted with human hepatocytes. As controls, uPA mice; heterozygous uPA$^{+/-}$-SCID mice not engrafted with human hepatocytes were used.

The rodent GAP PbΔ*b9*Δ*slarp* completely arrested early in liver-stage development in two different mouse strains after injection of very high number of sporozoites. In addition, immunizations with PbΔ*b9*Δ*slarp* efficiently induced sterile and long-lasting protective immunity in both BALB/c and C57BL/6 mice. Similarly, the Pf GAP PfΔ*slarp*Δ*b9* completely aborted development in human hepatocytes 1 day after invasion, while sporozoites were fully motile and invasive with infectivity comparable to Pf WT sporozoites. Importantly, asexual parasite growth and production of salivary gland sporozoites in the mosquito were unaffected ensuring normal GAP production. PbΔ*b9*Δ*slarp* is to our knowledge the first completely attenuated rodent mutant in which multiple genes have been deleted that are critical for two independent biological processes during liver-stage development, that is regulation of parasite genes/transcripts that play a role in early liver-stage development stages (*Silvie et al., 2008*; *Aly et al., 2011*) and the establishment of the PV within the infected hepatocyte (*Annoura et al., 2014*).

A number of Pb and Py GAPs have previously been reported to arrest at different time points during development in the liver (*Khan et al., 2012*; *Nganou-Makamdop and Sauerwein, 2013*). These include GAPs based on genes essential for i) the formation and maintenance of a parasitophorous vacuole (PV) (*b9*, *p52*, *p36*, *uis3*, and *uis4*; (*van Dijk et al., 2005*; *Kumar et al., 2009*; *Annoura et al., 2014*) and ii) type II fatty acid synthesis (i.e. *fabb/f*, *fabz*, *pdh e1α*; (*Vaughan et al., 2009*; *Annoura et al., 2012*)), and iii) the regulation of gene expression in the liver-stages (*sap1/slarp* (*Aly et al., 2008*; *Silvie et al., 2008*; *Aly et al., 2011*)). A critical safety requirement for GAPs in order to qualify as vaccine candidate is the total absence of blood infections during immunization and therefore the complete abrogation of liver-stage development. Unfortunately many of the above mentioned target genes including *p52*, *p36*, and those involved in type II fatty acid synthesis show a leaky phenotype, resulting in blood stage infections after administration of high number of sporozoites. Incomplete liver-stage arrest obviously disqualifies GAPs for further clinical development for safety reasons.

In *P. falciparum*, GAPs have been generated that lack both the *p52* and *p36* genes (*van Schaijk et al., 2008*; *VanBuskirk et al., 2009*). In the Pb rodent model, this GAP was not completely attenuated (*Annoura et al., 2012*). Similarly, this Pf GAP while severely attenuated by the lack of both genes, a low percentage of parasites of this GAP are able to develop into mature liver-stage (*Annoura et al., 2012*). These observations indicate a partially redundant function for these proteins; indeed, a breakthrough blood infection was observed in one out of the six volunteers after exposure to the bite of mosquitoes infected with sporozoites of a PfΔ*p52*Δ*p36* GAP (*Spring et al., 2013*).

Since functional redundancy of related genes has been reported more often in *Plasmodium* (*Liu et al., 2006*; *Heiss et al., 2008*; *van Dijk et al., 2010*; *Lin et al., 2013*), we pursued the generation of GAPs from which multiple genes were removed from the genome, each governing a critical yet independent cellular process. The selection of those target genes excluded type II fatty acid synthesis (FAS II) because *P. falciparum* mutants lacking FAS II enzymes fail to generate sporozoites inside the oocyst, indicating that the FAS II pathway is essential for sporogony (*van Schaijk et al., 2013*). The gene encoding liver-stage antigen 1 (LSA-1) may be an attractive candidate, but no orthologues are present in rodent or non-human primate *Plasmodium* species precluding sufficient pre-clinical testing (*Mikolajczak et al., 2011*). The reverse is true for two published rodent GAPs with deletions of the genes *uis3* or *uis4* of which unequivocal orthologues are absent in the *P. falciparum* genome.

Alternatively, genes encoding proteins with a role in the late stage parasite liver development could be an attractive target, since induction of protection by late arresting GAPs may be superior to early arresting GAPs (*Butler et al., 2011*; *Nganou-Makamdop and Sauerwein, 2013*) However, late arresting GAPs are likely more risky and prone to breakthrough infection as shown for GAPs lacking the genes *palm* or *lisp* (*Khan et al., 2012*).

Therefore, we decided to focus on early liver-stage arrest and selected the newly identified *b9* as a prime candidate. PbΔ*b9* elicits long-lived protective immune responses in mice and only few breakthrough blood infections occur in mice, albeit less than were observed with PbΔ*p52*Δ*p36* GAP sporozoites (*Annoura et al., 2012*). The genes *p52, p36,* and *b9,* all belong to the recently expanded 6-Cys family of *Plasmodium* proteins and may share a similar function in formation or maintenance of the PV membrane at the interface of parasite and host cell. Indeed, a triple gene-deletion mutant lacking *p52, p36,* and *b9* is no more attenuated than a mutant lacking *b9,* suggesting that these genes do not drive independent biological pathways (*van Dijk et al., 2005*; *Ploemen et al., 2012*; *Annoura et al., 2014*). To date, the early arresting *slarp* mutant is the only rodent GAP with a Pf ortholog without a record of breakthrough blood infections in mice. Indeed, our data confirm that rodent sporozoites lacking *slarp* are fully capable of hepatocyte invasion and formation of a PV but completely abort development soon after invasion as previously reported (*Aly et al., 2008*; *Silvie et al., 2008*; *Aly et al., 2011*). In this study, we report for the first time that *P. falciparum* mutants lacking *slarp,* that is PfΔ*slarp,* completely arrest at day 3 post-infection of primary human hepatocytes, while morphologically normal liver-stage parasites are still observed at 48 hpi. PfΔ*b9* parasites arrest at a point in time before day 2 after hepatocyte invasion, with the exception of a single liver schizont observed at a later time point (*Annoura et al., 2014*). The multiple attenuated PbΔ*b9*Δ*slarp* indeed passed our stringent pre-clinical safety screen and no breakthrough blood infections were observed in all conditions tested. In addition, we showed that immunization with PbΔ*b9*Δ*slarp* sporozoites induced strong and sustained protective immunity in BALB/c and C57BL/6 mice, with similar efficacy as reported for mutant sporozoites lacking P52 (or P52 and P36) or γ-radiated sporozoites (*Nussenzweig et al., 1967*; *van Dijk et al., 2005*; *Douradinha et al., 2007*; *Labaied et al., 2007*).

Live vaccine strains (attenuated by natural selection or genetic engineering) may be potentially released into the environment. Therefore, safety issues concerning the medical as well as environmental aspects must be considered including the absence of heterologous DNA sequences (in particular drug resistance genes) from the genome of GAPs (*Committee for Medical Products for Human Use, 2006*; *Frey, 2007*). Thus, a PfΔ*b9*Δ*slarp* GAP was generated free of a drug resistance marker using FRT/FLPe-recombinase methodology. This approach permits the removal of drug resistance markers that were introduced to generate the mutant and results in an altered genome that retains only two 34 nucleotide FRT sequences. The removal of the drug resistance marker has the additional advantage that these parasites are easily amenable to further genetic modification (*van Schaijk et al., 2010*).

The PfΔ*b9*Δ*slarp* GAP aborted early development in cultured primary human hepatocytes, with a phenotype and timing similar to PfΔ*b9,* and studies performed in a limited number of chimeric mice engrafted with human hepatocytes confirm this arrest phenotype. From the combined Pb and Pf data, one can conclude that Δ*b9* attenuation phenotype induces highly effective protection, although it may at a low frequency produce a breakthrough blood infection. Therefore, the additional deletion of *slarp* in these mutants provides these parasites with complete attenuation that is essential in order to proceed with human trials.

An important prerequisite for further downstream clinical development and manufacturing (*Seder et al., 2013*) is to show that production of PfΔ*b9*Δ*slarp* sporozoites is unabated and similar to WT parasites. We have shown that the PfΔ*b9*Δ*slarp* GAP produces WT numbers of sporozoites that are fully capable of infecting hepatocytes. In addition, we have produced aseptic, purified, cryopreserved PfΔ*b9*Δ*slarp* sporozoites (data not shown). Preliminary data from a 6-day attenuation assay in HC-04 cells showed that like irradiated PfSPZ (*Hoffman et al., 2010*; *Epstein et al., 2011*), none of the PfΔ*b9*Δ*slarp* sporozoites developed to mature liver-stage parasites expressing PfMSP-1 (data not shown), as aseptic, purified, cryopreserved WT sporozoites (*Roestenberg et al., 2013*).

In conclusion, we have generated a multiply attenuated PfΔ*b9*Δ*slarp* GAP, free of any drug resistance gene, and demonstrated that PfΔ*b9*Δ*slarp* sporozoites invade hepatocytes comparably to WT sporozoites and are completely attenuated. These findings provide a solid foundation for clinical development and testing of a PfSPZΔ*b9*Δ*slarp* vaccine.

### Note added at proof

While this manuscript was in preparation an article was published that also describes a multiple-gene deletion *P. falciparum* parasite that has undergone pre-clinical evaluation (*Mikolajczak et al., 2014*). In that study, the authors describe a *P. falciparum* mutant that, like our work, also lacks the gene *slarp* (*sap1*) as well as the paralogous pair of genes, p52 and p36.

## Materials and methods

### *P. berghei* reference parasite lines

The following reference lines of the ANKA strain of *P. berghei* were used: line cl15cy1 (*Janse et al., 2006a*, *2006b*) and line 676m1cl1 (*Pb*GFP-Luc$_{con}$; see RMgm-29 in www.pberghei.eu). *Pb*GFP-Luc$_{con}$ expresses a fusion protein of GFP and luciferase from the *eef1a* promoter (*Franke-Fayard et al., 2004*; *Janse et al., 2006a*).

### *P. falciparum* parasites and culture

For transfections, the parasite used was directly from a characterized good manufacturing process (GMP) and produced working cell bank (WCB) of the *P. falciparum* NF54 wild-type strain (*Ponnudurai et al., 1981*), produced by Sanaria Inc, identical to that described previously (*Hoffman et al., 2010*; *Epstein et al., 2011*; *Roestenberg et al., 2013*). Blood stages of wt, PfΔ*slarp*-a, PfΔ*slarp*-b, PfΔ*b9*Δ*slarp*-F7, and PfΔ*b9*Δ*slarp*-G9 were cultured in a semi-automated culture system using standard in vitro culture conditions for *P. falciparum* and induction of gametocyte production in these cultures was performed as previously described (*Ifediba and Vanderberg, 1981*; *Ponnudurai et al., 1982*, *1989*). Fresh human red blood cells and serum were obtained from Dutch National blood bank (Sanquin Nijmegen, NL; permission granted from donors for the use of blood products for malaria research). Cloning of transgenic parasites was performed by the method of limiting dilution in 96-well plates as described (*Thaithong, 1985*). Parasites of the positive wells were transferred to the semi-automated culture system and cultured for further phenotype and genotype analyses (See below).

### Experimental animals

For *P. berghei* infections, female C57BL/6J and BALB/c (12-week old; Janvier France) and Swiss OF1 (8 weeks old Charles River) were used. All animal experiments with rodent parasites performed at the LUMC (Netherlands) were approved by the Animal Experiments Committee of the Leiden University Medical Center (DEC 07171; DEC 10099) and at the RUNMC (Netherlands) by the Radboud University Experimental Animal Ethical Committee (RUDEC 2008-123, RUDEC 2008-148, RUDEC 2010-250, RUDEC 2011-022, RUDEC 2011-208). The Dutch Experiments on Animal Act is established under European guidelines (EU directive 86/609/CEE) regarding the Protection of Animals used for Experimental and Other Scientific Purposes.

Human liver-uPA-SCID mice (chimeric mice) were produced as described before (*Meuleman et al., 2005*). The study protocol for infecting these mice with *P. falciparum* sporozoites was approved by the animal ethics committee of the Faculty of Medicine and Health Sciences of the Ghent University.

### Generation and genotyping of *P. berghei* mutants

To disrupt the *P. berghei slarp* gene (PBANKA_090210), a construct was generated using the adapted 'Anchor-tagging' PCR-based method as described (*Annoura et al., 2012*) (*Figure 1—figure supplement 1*). The two targeting fragments (1195 bp and 823 bp) of *slarp* were amplified using genomic DNA (parasite line cl15cy1) as template with the primer pairs 5960/5961 (5′target sequence) and 5962/5963 (3′target sequence). See *Supplementary file 2A* for the sequence of the primers. Using this PCR-based targeting construct (pL1740), the mutant PbΔ*slarp*-a (1839cl3) was generated in the *Pb*GFP-Luc$_{con}$ reference line using standard methods of transfection and positive selection with pyrimethamine (*Figure 1—figure supplement 1*). The generation of the drug-selectable marker-free mutant PbΔ*b9*Δ*sm* (1309cl1m0cl2; RMgmDB no. 934) has been described by *Annoura et al. (2014)*. This mutant, which contains a disrupted *b9* gene and is drug-selectable marker free, was used for deleting the *slarp* gene (PBANKA_090210). To delete the *slarp* gene, the gene-deletion construct pL1740 was used as described above. Using this construct the mutant PbΔ*b9*Δ*slarp* (line 1844cl1) was generated in the PbΔ*b9*Δ*sm* line using standard methods of transfection and positive selection with pyrimethamine (*Figure 1*).

Correct integration of the constructs into the genome of mutant parasites was analyzed by diagnostic PCR-analysis and Southern analysis of PFG-separated chromosomes as shown in *Figure 1* and *Figure 1—figure supplement 1*. PFG-separated chromosomes were hybridized with a probe recognizing *hdhfr* or the 3′-UTR *dhfr/ts* of *P. berghei* (*Janse et al., 2006b*).

## Generation and genotyping of *P. falciparum* mutants

The *slarp* gene (PF3D7_1147000) in *P. falciparum* WT parasites (NF54wcb) was deleted using a modified construct based on plasmid pHHT-FRT-(GFP)-Pf52 (*van Schaijk et al., 2010*) (*Figure 2—figure supplement 1*). Targeting regions were generated by PCR using primers BVS179 and BVS180 for the 5′ target region and primers BVS182 and BVS184 for the 3′ target region (see *Supplementary file 2B* for primer sequences). The 5′and 3′ target regions were cloned into pHHT-FRT-(GFP)-Pf52 digested with *Bsi*WI, *Bss*HII and *Nco*I, *Xma*I, respectively, resulting in the plasmid pHHT-FRT-GFP-*slarp*. The *b9* gene (PF3D7_0317100) of PfΔ*slarp*-b *P. falciparum* parasites was deleted using a modified construct based on plasmid pHHT-FRT-(GFP)-Pf52 (*van Schaijk et al., 2010*) (*Figure 2—figure supplement 1*). Targeting regions were generated by PCR using primers BVS84 and BVS85 for the 5′ target region and primers BVS88 and BVS89 for the 3′ target region. The 5′and 3′ target regions were cloned into pHHT-FRT-(GFP)-Pf52 digested with *Nco*I, *Xma*I and *Mlu*I, *Bss*HII resulting in the plasmid pHHT-FRT-GFP-*b9*. All DNA fragments were amplified by PCR amplification (Phusion, Finnzymes) from genomic *P. falciparum* DNA (NF54 strain) and all PCR fragments were sequenced after TOPO TA (Invitrogen, Leek, The Netherlands) sub-cloning. Transfection of WT (NF54wcb) parasites with the plasmid pHHT-FRT-GFP-*slarp* and selection of mutant parasites were performed, as described (*van Schaijk et al., 2010*), resulting in the selection of the parasite line PfΔ*slarp*-a. The second PfΔ*slarp* parasite line, originating from an independent transfection, was subsequently transfected with pMV-FLPe to remove the drug-selectable marker cassette using FLPe as described (*van Schaijk et al., 2010*) and cloned resulting in the parasite clone PfΔ*slarp*-b. Subsequent transfection of PfΔ*slarp*-b parasites with the plasmid pHHT-FRT-GFP-*b9* and selection were performed, as described above, resulting in the parasite line PfΔ*b9*Δ*slarp*. The parasite line PfΔ*b9*Δ*slarp* was subsequently transfected with pMV-FLPe to remove the drug-selectable marker cassette using FLPe and cloned, as described above, resulting in the cloned parasite lines PfΔ*b9*Δ*slarp*-F7 and PfΔ*b9*Δ*slarp*-G9 that are free of drug resistance markers.

Genotype analysis of PfΔ*slarp* and PfΔ*b9*Δ*slarp* parasites was performed by Expand Long range dNTPack (Roche) diagnostic, long-range, PCR (LR-PCR) and Southern blot analysis (*Figure 2—figure supplement 2*). Genomic DNA of blood stages of WT or mutant parasites was isolated and analyzed by LR-PCR using primer pair p1, p2 (*slarp*) and p3, p4 (*b9*) (See *Supplementary file 2B* for primer sequences) for correct integration of the constructs in the respective *slarp* and *b9* loci by double cross-over homologous recombination. The LR-PCR program has an annealing step of 48°C for 30 s and an elongation step of 62°C for 10–15 min. All other PCR settings were according to manufacturer's instructions. PCR products were directly analyzed by standard agarose gel electrophoresis or first digested with restriction enzymes for further confirmation of the genotype and removal of resistance markers was confirmed by sequencing. For Southern blot analysis, genomic DNA was digested with *Taq*I or *Rca*I restriction enzymes for analysis of integration into the *slarp* and *b9* loci, respectively. Southern blot was generated by capillary transfer as described (*Sambrook and Russel, 2001*) and DNA was hybridized to radioactive probes specific for the targeting regions used for the generation of the mutants and generated by PCR (See above).

The presence or absence of *slarp* and *b9* transcripts in WT and mutant sporozoites was analyzed by reverse transcriptase-PCR (*Figure 2—figure supplement 2*). Total RNA was isolated using the RNeasy mini Kit (Qiagen) from $10^6$ salivary gland sporozoites collected by dissection of mosquitoes 16 days after feeding with WT, PfΔ*slarp*-a, PfΔ*slarp*-b, PfΔ*b9*Δ*slarp*-F7, and PfΔ*b9*Δ*slarp*-G9 parasites. Remaining DNA was degraded using DNAseI (Invitrogen). cDNA was synthesized using the First Strand cDNA synthesis Kit for RT-PCR AMV (Roche). As a negative control for the presence of genomic DNA, reactions were performed without reverse transcriptase (RT−). PCR amplification was performed for regions of *slarp* using primers BVS290, BVS292 and for regions of *b9* using primers BVS286 and BVS288. Positive control was performed by PCR of 18S rRNA using primers 18Sf and 18Sr.

## Phenotype analyses of blood stages of *P. berghei* and *P. falciparum* mutants

Asexual multiplication rate and gametocyte production of *P. berghei* blood stages were determined as described (*Annoura et al., 2012*). The *P. berghei* mutants were maintained in Swiss mice. The

multiplication rate of blood stages and gametocyte production were determined during the cloning procedure (*Janse et al., 2006b*) and were not different from parasites of the reference ANKA lines. *P. falciparum* blood stage development and gametocyte production were analyzed as described (*van Schaijk et al., 2010*).

## Analysis of *P. berghei* and *P. falciparum* sporozoite production and in vitro motility, hepatocyte traversal, and infectivity of sporozoites

Feeding of *A. stephensi* mosquitoes with *P. berghei* and *P. falciparum*, determination of oocyst production and sporozoite collection, as well as *P. berghei* gliding motility were performed as described (*Annoura et al., 2012*). *P. falciparum* gliding motility of sporozoites was determined as described (*Stewart and Vanderberg, 1988*; *van Schaijk et al., 2008*). *P. falciparum* cell traversal and invasion of hepatocytes were determined in Huh7 cells and primary human hepatocytes respectively as described (*van Schaijk et al., 2008*). Infectivity of *P. berghei* sporozoites and development was determined in cultures of Huh7 cells as described (*van Schaijk et al., 2008*). For analysis of liver-stage development by immunofluorescence, parasites were stained with the following primary antibodies: anti-PbEXP1 (PBANKA_092670; raised in chicken (*Sturm and Heussler, 2007*)) and anti-PbHSP70 (PBANKA_081890; raised in mouse (*Mueller et al., 2005*)). Infectivity of *P. falciparum* sporozoites and development was analyzed in primary human hepatocytes as described (*van Schaijk et al., 2008*). Briefly for analysis of development by immunofluorescence, parasites were stained with the following primary antibodies: anti-HSP70 (PF3D7_0930300 (*Renia et al., 1990*)) and anti-CSP (PF3D7_0304600; 3SP2) using double labeling. Anti-mouse secondary antibodies, conjugated to Alexa-488 or Alexa-594 (Invitrogen), were used for visualization. Primary human hepatocytes were isolated from healthy parts of human liver fragments, which were collected during unrelated surgery in agreement with French national ethical regulations (*Gouagna et al., 2007*) and after oral informed consent from adult patients undergoing partial hepatectomy as part of their medical treatment (Service de Chirurgie Digestive, Hépato-Bilio-Pancréatique et Transplantation Hépatique, Hôpital Pitié-Salpêtrière, Paris, France). The collection and use of this material for the purposes of the study presented here were undertaken in accordance with French national ethical guidelines under Article L. 1121-1 of the 'Code de la Santé Publique'. Given that the tissue samples are classed as surgical waste, that they were used anonymously (the patient's identity is inaccessible to the researchers), and that they were not in any way genetically manipulated, article L. 1211-2 stipulates that their use for research purposes is allowed provided that the patient does not express any opposition to the surgeon prior to surgery and after being informed of the nature of the research in which they might be potentially employed. Within this framework, the collection and use of this material was furthermore approved by the Institutional Review Board (Comité de Protection des Personnes) of the Centre Hospitalo-Universitaire Pitié-Salpêtrière, Assistance Publique-Hôpitaux de Paris, France.

## Analysis of *P. berghei* sporozoite infectivity in mice and in vivo imaging of liver-stage development

C57BL/6 or BALB/c mice were inoculated with sporozoites by intravenous injection of different sporozoite numbers, ranging from $1 \times 10^4$ to $5 \times 10^5$. Blood stage infections were monitored by analysis of Giemsa-stained thin smears of tail blood collected on day 4–14 after inoculation of sporozoites. The prepatent period (measured in days post sporozoite infection) is defined as the day when a blood stage infection with a parasitemia of 0.5–2% is observed. Liver-stage development in live mice was monitored by real time in vivo imaging of liver-stages as described (*Ploemen et al., 2012*). Liver-stages were visualized by measuring luciferase activity of parasites (expressing luciferase under the *eef1a* promoter) in whole bodies of mice (*Ploemen et al., 2009*).

### Immunizations of mice with *P. berghei* sporozoites

Prior to immunization, *P. berghei* sporozoites were collected at day 21–27 after mosquito infection by hand-dissection. Salivary glands were collected in DMEM (Dulbecco's Modified Eagle Medium from GIBCO) and homogenized in a homemade glass grinder. The number of sporozoites was determined by counting in triplicate in a Bürker-Türk counting chamber using phase-contrast microscopy. BALB/c and C57BL/6 mice were immunized by intravenous injection using different numbers of mutant sporozoites. BALB/c mice received one immunization and C57BL/6 mice received three immunizations with two 7 day intervals. Immunized mice were monitored for blood infections by analysis of Giemsa stained films of tail blood at day 4–16 after immunization. Immunized mice were challenged at different time

points after immunization by intravenous injection of $1 \times 10^4$ sporozoites from the *P. berghei* ANKA reference line cl15cy1. In each experiment, age matched naive mice were included to verify infectivity of the sporozoites used for challenge. After challenge, mice were monitored for blood infections by analysis of Giemsa stained films of tail blood at day 4–21.

## Development of Pf *Δb9Δslarp* GAP in chimeric mice engrafted with human hepatocytes

Human liver-uPA-SCID mice were produced as described before (*Meuleman et al., 2005*). Briefly, within two weeks after birth homozygous uPA$^{+/+}$-SCID mice (*Foquet et al., 2013*) were transplanted with approximately $10^6$ cryopreserved primary human hepatocytes obtained from a single donor (BD Biosciences, Erembodegem, Belgium). To evaluate successful engraftment, human albumin was quantified in mouse plasma with an in-house ELISA (Bethyl Laboratories Inc., Montgomery, TX). The study protocol was approved by the animal ethics committee of the Faculty of Medicine and Health Sciences of the Ghent University. Human liver-uPA-SCID mice (n = 10) and non-chimeric heterozygous uPA$^{+/-}$-SCID mice (control, n = 2) were intravenously injected with $10^6$ fresh isolated PfΔb9Δslarp-G9 or as a control WT sporozoites. One and 5 days post-infection livers were removed and each liver was cut into 12 standardized sections and stored in RNAlater (Sigma) at 4°C until analysis as described (*Foquet et al., 2013*). From each part DNA was extracted to assess the parasite load by Pf18S qPCR and to assess the number of human and mouse hepatocytes by Multiplex qPCR PTGER2 analysis (*Foquet et al., 2013*).

While this manuscript was in preparation an article was published that also describes a multiple-gene deletion *P. falciparum* parasite that has undergone pre-clinical evaluation (*Mikolajczak et al, 2014*). In that study, the authors describe a *P. falciparum* mutant that, like our work, also lacks the gene *slarp* (*sap1*) as well as the paralogous pair of genes, p52 and p36.

## Data and Materials availability

The materials described in this study must be acquired through a material transfer agreement.

## Acknowledgements

We would like to thank the following people from RUMC (Nijmegen) for technical support: Claudia Lagarde, Alex Ignacio, Daniëlle Janssen, Rianne Siebelink-Stoter, Wouter Graumans, Jolanda Klaassen, Laura Pelser-Posthumus, Astrid Pouwelsen, and Jacqueline Kuhnen; and the following people from LUMC (Leiden) for technical support: Jai Ramesar, Jing-wen Lin, and Hans Kroeze. We acknowledge the Sanaria Manufacturing Team for the GMP produced working cell bank of PfNF54. Sanaria's development efforts supported in part by NIAID Small Business Innovation Research grants, 5R44AI069631 and 5R44AI058375.

## Additional information

### Competing interests

SLH: CEO of Sanaria Inc, biotechnology company focused on whole sporozoite malaria vaccines. The other authors declare that no competing interests exist.

### Funding

| Funder | Grant reference number | Author |
|---|---|---|
| Top Institute Pharma (TI Pharma) | T4-102 | Ben C L van Schaijk, Ivo H J Ploemen, Takeshi Annoura, Martijn W Vos, Geert-Jan van Gemert, Severine Chevalley-Maurel, Marga van de Vegte-Bolmer, Mohammed Sajid, Cornelius C Hermsen, Stephen L Hoffman, Chris J Janse, Robert W Sauerwein |

The funder had no role in study design, data collection and interpretation, or the decision to submit the work for publication.

### Author contributions

BCLS, IHJP, Conception and design, Acquisition of data, Analysis and interpretation of data, Drafting or revising the article; TA, Conception and design, Acquisition of data, Analysis and interpretation

of data; MWV, Acquisition of data, Analysis and interpretation of data; LF, Conception and design, Acquisition of data; G-JG, SC-M, MV-B, MS, J-FF, AL, Acquisition of data, Analysis and interpretation of data, Drafting or revising the article; GL-R, PM, Analysis and interpretation of data, Drafting or revising the article, Contributed unpublished essential data or reagents; CCH, Analysis and interpretation of data, Drafting or revising the article; DM, CJJ, SMK, Conception and design, Analysis and interpretation of data, Drafting or revising the article; SLH, Conception and design, Drafting or revising the article, Contributed unpublished essential data or reagents; RWS, Conception and design, Drafting or revising the article

### Ethics

Human subjects: Primary human hepatocytes were isolated from healthy parts of human liver fragments which were collected during unrelated surgery in agreement with French national ethical regulations and after oral informed consent from adult patients undergoing partial hepatectomy as part of their medical treatment (Service de Chirurgie Digestive, Hépato-Bilio-Pancréatique et Transplantation Hépatique, Hôpital Pitié-Salpêtrière, Paris, France). The collection and use of this material for the purposes of the study presented here were undertaken in accordance with French national ethical guidelines under Article L. 1121-1 and article L. 1211-2.

Animal experimentation: All animal experiments with rodent parasites performed at the LUMC (Netherlands) were approved by the Animal Experiments Committee of the Leiden University Medical Center (DEC 07171; DEC 10099) and at the RUNMC (Netherlands) by the Radboud University Experimental Animal Ethical Committee (RUDEC 2008-123, RUDEC 2008-148, RUDEC 2010-250, RUDEC 2011-022, RUDEC 2011-208). The Dutch Experiments on Animal Act is established under European guidelines (EU directive 86/609/CEE) regarding the Protection of Animals used for Experimental and Other Scientific Purposes. Human liver-uPA-SCID mice (chimeric mice) were produced as described before. The study protocol for infecting these mice with *P. falciparum* sporozoites was approved by the animal ethics committee of the Faculty of Medicine and Health Sciences of the Ghent University. The study protocol was approved by the animal ethics committee of the Faculty of Medicine and Health Sciences of the Ghent University.

---

## Additional files

### Supplementary files

• Supplementary file 1. Oocyst and sporozoite production and sporozoite characteristics (motility, traversal, hepatocyte invasion) of the *P. berghei* mutants PbΔ*slarp* and PbΔ*b9*Δ*slarp*.

• Supplementary file 2. Primer sequences.

---

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
