## [Decision Letter]

Thank you for sending your work entitled “A genetically attenuated malaria vaccine candidate based on *P. falciparum* b9/slarp gene-deficient sporozoites” for consideration at *eLife*. Your article has been favorably evaluated by Prabhat Jha (Senior editor), Nicholas White (Reviewing editor), and 3 reviewers.

The Reviewing editor and the reviewers discussed their comments before we reached this decision, and the Reviewing editor has assembled the following comments to help you prepare a revised submission.

The paper describes an important step on the pathway to the development of a genetically attenuated parasite (GAP) for use as a live-attenuated vaccine for human malaria.

1) Could you explain why you have focused on Δb9, given that it suffers from the same 'redundancy' issues affecting previous liver-stage genes examined in GAP mutants? Why did you not focus on slarp until another truly 'essential' liver-stage gene is found for combination? The b9 deletion seems to have limited value.

2) Are you sure that the Pb double KO parasites did not give rise to sub-patent infections at densities only detectable by PCR?

3) Please provide numbers of sporozoites added to the human hepatocyte line for the in vitro infectivity study (Figure 2).

4) Only 5 humanized mice were studied for the *in vivo* infectivity study, with one becoming PCR positive above background at day 1 and 0 of 3 above background at day 5. These data are encouraging but don't exclude the possibility that a rare parasite might break through. The limitations of the small numbers should be acknowledged.

---

## [Author Response]

*1) Could you explain why you have focused on Δb9, given that it suffers from the same 'redundancy' issues affecting previous liver-stage genes examined in GAP mutants? Why did you not focus on slarp until another truly 'essential' liver-stage gene is found for combination? The b9 deletion seems to have limited value*.

First of all it was considered to be essential to delete two functionally independent genes to minimize the chances of breakthrough infections based on experiences in the past with single gene deletions. It is critical that GAPs are incapable of ‘reverting’ to a wild-type mode of (blood stage) development, and this is best achieved by removing multiple genes from the parasite’s genome. In that perspective, slarp was considered the best partner gene to ensure the safety profile as best and only the single gene deletion able to generate completely liver-stage arrested parasites in multiple *Plasmodium* species to date. Subsequently the b9 gene has been selected while comparing other gene candidates as best target for the following reasons:

(1) Δ*b9* GAP are strongly attenuated as only very high doses of Δ*b9* sporozoites (>50 000) can produce a blood stage infection, in 1 of 3 mouse strains examined and only a very low percentage of these mice become infected even at very high sporozoite inocula (i.e. 500 000); the Δ*b9* GAP has the strongest attenuation of all reported single gene GAPs, after Δ*slarp*-GAP, and is considerably more attenuated than parasites that lack BOTH *p52* and *p36* (the next best GAP candidates)

(2) *b9* encodes for a protein that is involved in a completely different biological process from SLARP, thereby any ‘reversion’ to wild type phenotype would require a very unlikely functional compensation of 2 independent cellular process;

(3) Δ*b9* GAP are most potent in induction of protective immunity in several murine models.

Taken together the combination of Δ*slarp* with Δ*b9* should result in a GAP that is both safe and more potent than any other combination currently available.

2) Are you sure that the Pb double KO parasites did not give rise to sub-patent infections at densities only detectable by PCR?

Indeed we have not analysed the blood of immunized and challenged mice by PCR or blood transfer to naive mice. Notwithstanding, we do not believe that sub-patent infections may occur or persist.

Δ*b9*Δ*slarp* blood stage parasite produces a normal course of infection reaching a parasitemia of 0.2-5% parasitemia on the same day similar to wild-type parasites. Therefore any Δ*b9*Δ*slarp* parasite that would emerge in the blood after liver stage development, in an unimmunised mouse, will become patent.

Extensive blood smear examinations in a large number of mice after single dose infections, has never provided any evidence for a sub-patent infection. Indeed many GAP immunized mice were housed for longer than 1 year after immunisation and some establish a normal blood stage infection after wild-type challenge.

We therefore believe it is unlikely that a sub-patent Δ*slarp*Δ*b9* blood stage infection is possible or present.

*3) Please provide numbers of sporozoites added to the human hepatocyte line for the in vitro infectivity study (*Figure 2*).*

The number of 40.000 sporozoites is now indicated in the results section and reads:

“PfΔ*b9*Δ*slarp* sporozoites showed normal gliding motility, hepatic cell traversal (Figure 1) as well as invasion of primary human hepatocytes, but parasites were completely absent in two independent experiments at day 2 up to day 7 post-infection following inoculation of primary human hepatocytes with 40.000 PfΔ*b9*Δ*slarp* F7 or G9 sporozoites (Figure 2)”

Additionally the amount of sporozoites is indicated in the figure legend of Figure 2 and now reads:

“C. Development of *P. falciparum* wt, PfΔ*slarp-*a PfΔ*slarp-*b (top panel), PfΔ*b9*Δ*slarp*-F7 and PfΔ*b9*Δ*slarp*-G9 (bottom panel) liver-stages in primary human hepatocytes following inoculation with 40.000 sporozoites. From day 2 to 7 the mean number ± standard deviation of parasites per 96-well was determined by counting parasites stained with anti-*P. falciparum* HSP70 antibodies. The bottom panel represents experiments performed in primary human hepatocytes from 2 different donors. No parasites present (NP).”

*4) Only 5 humanized mice were studied for the in vivo infectivity study, with one becoming PCR positive above background at day 1 and 0 of 3 above background at day 5. These data are encouraging but don't exclude the possibility that a rare parasite might break through. The limitations of the small numbers should be acknowledged*.

We agree with the reviewers that based on the limited number of mice that we used we cannot exclude the possibility of a breakthrough infections in humanised mice. However, in all our studies performed with both the single slarp deletion and the double slarp/b9 deletion mutants we were unable to detect developing parasites in cultured primary human hepatocytes. In order to address the reviewers concern we have now modified sentences in the Results section as follows:

“Although these studies were performed with a limited number of mice, these findings indicate that PfΔb9Δslarp parasites can invade but do not develop in livers of humanized mice. Our combined results demonstrate abrogation of development of PfΔb9Δslarp inside human hepatocytes.”

And in the Discussion section:

“The PfΔb9Δslarp GAP completely aborted early development in cultured primary human hepatocytes with a phenotype and timing similar to PfΔb9 and studies performed in a limited number of chimeric mice engrafted with human hepatocytes also confirm this arrest phenotype.”